# The Expression Profiles of lncRNAs Are Associated with Neoadjuvant Chemotherapy Resistance in Locally Advanced, Luminal B-Type Breast Cancer

**DOI:** 10.3390/ijms25158077

**Published:** 2024-07-24

**Authors:** Miguel González-Woge, Laura Contreras-Espinosa, José Antonio García-Gordillo, Sergio Aguilar-Villanueva, Enrique Bargallo-Rocha, Paula Cabrera-Galeana, Tania Vasquez-Mata, Ximena Cervantes-López, Diana Sofía Vargas-Lías, Rogelio Montiel-Manríquez, Luis Bautista-Hinojosa, Rosa Rebollar-Vega, Clementina Castro-Hernández, Rosa María Álvarez-Gómez, Inti Alberto De La Rosa-Velázquez, José Díaz-Chávez, Francisco Jiménez-Trejo, Cristian Arriaga-Canon, Luis Alonso Herrera

**Affiliations:** 1Unidad de Investigación Biomédica en Cáncer, Instituto Nacional de Cancerología-Instituto de Investigaciones Biomédicas, Universidad Nacional Autónoma de México, Avenida San Fernando No. 22 Col. Sección XVI, Tlalpan, Mexico City C. P. 14080, Mexico; mgw996@gmail.com (M.G.-W.); lauram.con.es@gmail.com (L.C.-E.); tvasquez491@gmail.com (T.V.-M.); ximecervlo04@ciencias.unam.mx (X.C.-L.); rogelio_montiel@hotmail.com (R.M.-M.); ccastroh7@yahoo.com.mx (C.C.-H.); jdiazchavez03@gmail.com (J.D.-C.); 2Posgrado en Ciencias Biológicas, Universidad Nacional Autónoma de México, Unidad de Posgrado, Edificio D, 1° Piso, Circuito de Posgrados, Ciudad Universitaria, Coyoacán, Mexico City C. P. 04510, Mexico; luisebh1919@gmail.com; 3Departamento de Oncología Médica de Mama, Instituto Nacional de Cancerología, Tlalpan, Mexico City C. P. 14080, Mexico; joseantonio.garciagordillo@gmail.com (J.A.G.-G.); drapaulacabrera@gmail.com (P.C.-G.); 4Departamento de Tumores Mamarios, Instituto Nacional de Cancerología, Avenida San Fernando No. 22 Col. Sección XVI, Tlalpan, Mexico City C. P. 14080, Mexico; sergioaguilarmd@gmail.com (S.A.-V.); enrique.bargallo@gmail.com (E.B.-R.); sofia.vargas.lias@gmail.com (D.S.V.-L.); 5Genomics Laboratory, Red de Apoyo a la Investigación, Universidad Nacional Autónoma de México, Tlalpan, Mexico City C. P. 14080, Mexico; rebollar@cic.unam.mx; 6Clínica de Cáncer Hereditario, Instituto Nacional de Cancerología, Avenida San Fernando No. 22 Col. Sección XVI, Tlalpan, Mexico City C. P. 14080, Mexico; rosamag2@hotmail.com; 7Genomics Core Facility, Helmholtz Zentrum Muenchen, Ingolstaedter Landstr 1, 85754 Neuherberg, Germany; inti.delarosavelazquez@helmholtz-munich.de; 8Tecnológico de Monterrey, Escuela de Medicina y Ciencias de la Salud, Monterrey C. P. 64710, Mexico; 9Instituto Nacional de Pediatría, Insurgentes Sur No. 3700-C, Coyoacán, Mexico City C. P. 04530, Mexico; trejofj@hotmail.com

**Keywords:** lncRNA, luminal B, breast cancer, transcriptome, predictive biomarker, neoadjuvant chemotherapy

## Abstract

lncRNAs are noncoding transcripts with tissue and cancer specificity. Particularly, in breast cancer, lncRNAs exhibit subtype-specific expression; they are particularly upregulated in luminal tumors. However, no gene signature-based laboratory tests have been developed for luminal breast cancer identification or the differential diagnosis of luminal tumors, since no luminal A- or B-specific genes have been identified. Particularly, luminal B patients are of clinical interest, since they have the most variable response to neoadjuvant treatment; thus, it is necessary to develop diagnostic and predictive biomarkers for these patients to optimize treatment decision-making and improve treatment quality. In this study, we analyzed the lncRNA expression profiles of breast cancer cell lines and patient tumor samples from RNA-Seq data to identify an lncRNA signature specific for luminal phenotypes. We identified an lncRNA signature consisting of LINC01016, GATA3-AS1, MAPT-IT1, and DSCAM-AS1 that exhibits luminal subtype-specific expression; among these lncRNAs, GATA3-AS1 is associated with the presence of residual disease (Wilcoxon test, *p* < 0.05), which is related to neoadjuvant chemotherapy resistance in luminal B breast cancer patients. Furthermore, analysis of GATA3-AS1 expression using RNA in situ hybridization (RNA ISH) demonstrated that this lncRNA is detectable in histological slides. Similar to estrogen receptors and Ki67, both commonly detected biomarkers, GATA3-AS1 proves to be a suitable predictive biomarker for clinical application in breast cancer laboratory tests.

## 1. Introduction

Breast cancer (BC) is the second most frequently diagnosed malignant neoplasm, occupying 23.8% of the newly diagnosed cancers per million women and 4.3 million deaths in 2022 [1,2]. Particularly in Mexico and other middle-income countries, between 30 and 60% of the new cases are locally advanced [3]. Locally advanced breast cancer (LABC) comprises a heterogeneous group of tumors with a high risk of local recurrence, and is generally associated with shorter disease-free survival (DFS) and overall survival (OS) than early-stage tumors [4]. For this group of patients, the therapy of choice is neoadjuvant chemotherapy (NAC), which has variable efficacy [5]. In clinical practice, breast tumors are classified into subtypes based on the detection of hormone receptor expression by immunohistochemistry [6] and molecular subtypes based on gene expression patterns [7]. Among these, hormone receptor-positive (HR+) luminal tumors are further divided into two distinct subtypes, luminal A and luminal B, each with unique molecular, pathologic, and clinical characteristics. Clinically, the luminal B subtype includes HR+ tumors (ER and/or PR) with high Ki67 expression (>14% or 30%) and variable HER2 expression (HER2 positive or negative), which is associated with worse recurrence-free and disease-specific survival [8,9]. Given that this subtype has a different incidence, response to treatment, and survival, it is paramount to explore prognostic and/or predictive biomarkers [10,11,12,13].

The current standard surrogate of event free survival after NAC is pathologic complete response (pCR) [14], which is defined as the absence of residual invasive cancer in breast tissue and lymph nodes after a complete course of NAC [15], and according to pCR status, patients can be classified as responders (patients who achieve pCR, i.e., ypT0/Tis, ypN0 in the AJCC TNM system) or nonresponders (patients with residual disease, RD). Even in the setting that it might not be an adequate surrogate of overall survival in HR+HER2- patients [14], recent real-world evidence has identified that the overall survival of the pCR population of Her2 positive, triple negative, and luminal B breast cancer is similar. It is also known that compared with luminal A, luminal B is nearly thirty times more likely to achieve pCR [14]. This subtype can have progression rates as high as 28% during NAC, making it a subtype of particular interest [16]. This is due to the presence of cell clones in residual tumors, which are considered to be resistant to the exposure treatment [17]. This scenario allows us to consider the intensification of postneoadjuvant treatment by implementing new therapeutic strategies, such as the use of cyclin 4 and 6 inhibitors [18]. However, the discrepancy in success rates after NAC can lead to overtreatment, which is associated with short-, mid-, and long-term toxicity; increased healthcare costs and unnecessary burdens on healthcare systems [19]. This has led to the development of a comprehensive approach to classify and treat LABC, as well as monitor disease progression and treatment response.

Although some biomarkers have been proposed and show promise for predicting the efficacy of NAC [20,21], most of these biomarkers are currently under development and are not ready for routine clinical use [21]. Therefore, a biomarker or set of biomarkers with the ability to predict the response to NAC could guide treatment by identifying patients who will benefit from this type of treatment. Gene expression panels, such as Oncotype Dx [22] and Mammaprint [23], use protein-coding genes to identify survival benefit when exposed to adjuvant chemotherapy [24]. However, only a few alternative parameters, such as Ki67 [25] and tumor-infiltrating lymphocytes (TILs) [26], have been proposed to predict the response to NAC. Recently, it has been reported that in addition to protein-coding genes, noncoding transcripts, such as long noncoding RNAs (lncRNAs), may be useful molecular markers for BC diagnosis and prognosis evaluation [27,28]. By definition, these transcripts have more than 200 bases and lack open reading frames, and are therefore not translated into proteins [29].

Recent research efforts using transcriptome analysis have revealed several clinically relevant lncRNAs in BC, such as DSCAM-AS1, a luminal subtype-specific transcript that regulates cell proliferation and invasion and is associated with tamoxifen resistance [30,31]. Other examples of clinically relevant lncRNAs in BC include MALAT1, whose overexpression promotes the upregulation of the following genes associated with metastasis [32]: LINK-A, which has been shown to contribute to immune evasion [33]; and UCA1, which is associated with resistance to trastuzumab [34]. In addition, the overexpression of the lncRNAs ROR [35], H19 [36], and MAPT-AS1 [37] has been linked to resistance to paclitaxel, one of the components of the NAC regimen for LABC [38]. Furthermore, the lncRNA GATA3-AS1 has been described as a potential biomarker of NAC response in luminal B breast cancer patients, and it has been identified using RNA-Seq transcriptome analysis and validated using RT-qPCR [27]. Taken together, these data demonstrate the clinical utility of lncRNAs identified using transcriptome analysis as predictive biomarkers in luminal breast cancer patients.

In this study, publicly available RNA-Seq data from Gene Expression Omnibus (GEO) datasets obtained from two international cohorts were reanalyzed to determine the expression profiles of lncRNAs in different molecular subtypes of BC. Additionally, RNA-Seq data from Hispanic breast cancer patients were used to validate these expression profiles. Subsequently, differential lncRNA expression between responders and nonresponders to NAC treatment led to the identification of candidate lncRNAs such as FAM222A-AS1, MAPT-IT1, and GATA3-AS1. Particularly, GATA3-AS1 showed its utility in predicting resistance to NAC, which was experimentally validated in pretreatment luminal B LABC formalin-fixed paraffin-embedded (FFPE) biopsies using RNA-ISH.

## 2. Results

### 2.1. Transcriptomic Profiling of lncRNAs in Breast Cancer Cell Lines

To determine the transcription profile of lncRNAs in the different breast cancer subtypes, we performed a transcriptome analysis with publicly available RNA-Seq data from 150 BC cell lines, which is the proposed approach for the discovery of molecular biomarkers from transcriptomic data (Appendix A). Clustering analysis was performed to classify and filter cell line data according to their gene expression profiles, which allowed for the selection of 42 BC cell lines (5 normal cell lines and 37 neoplastic cell lines, including 10 luminal A, 8 luminal B, 9 HER2-enriched, and 10 triple-negative cell lines). To determine the luminal B-specific lncRNA expression profile, two different bioinformatic workflows were used, followed by differential expression gene analysis in malignant luminal BC cell lines. Then, the differentially expressed lncRNAs that were identified using both pipelines were selected (Figure 1).

Transcriptome analysis of 42 BC cell lines revealed 2874 differentially expressed genes in BC malignant cell lines; among these genes, 585 mRNAs, 199 lncRNAs, and 45 transcripts with other functions (such as pseudogenes and snoRNAs) were downregulated, and 1546 mRNAs and 499 lncRNAs, including the lncRNA HOTAIR, the BC-specific lncRNA LINC01016 [39], AL157387.1, the luminal subtype-related lncRNAs DSCAM-AS1 [30], and GATA3-AS1 [27] were upregulated (Figure 2A). A Gene set enrichment analysis (GSEA) of these differentially expressed genes showed their association with biological processes involved in mammary tumor development, such as cell differentiation, cell migration, and extracellular matrix organization (Appendix A).

Furthermore, from of the 42 breast cancer (BC) cell lines evaluated, 18 were characterized as luminal BC cell lines. To investigate the existence of a luminal subtype-specific lncRNA expression profile, the results of the differential expression analysis between malignant and nonmalignant BC cell lines were used to construct a nonhierarchical clustering heatmap of the top 40 lncRNAs differentially expressed in luminal tumors. This analysis revealed that eleven lncRNAs, including GATA3-AS1 and LINC01016, have luminal subtype-specific expression (Figure 2B). The lncRNA GATA3-AS1 has previously been implicated in resistance to NAC [27], while LINC01016 has been shown to be specifically expressed in estrogen receptor-positive breast tumors [39]. In summary, these results indicate the existence of a potential luminal breast cancer-specific expression profile of lncRNAs.

Although hormone receptors are commonly expressed in luminal tumors, and luminal B tumors are diagnosed at a considerable frequency in clinical practice (approximately 35%), these tumors present variable outcomes to NAC treatment; only 16% of patients achieve a pCR [40]. Thus, it is necessary to identify molecular biomarkers that could have predictive value for NAC response in luminal B breast cancer patients. Thus, differential expression analysis was performed to assess the expression profiles of lncRNAs associated with the luminal subtype B, which revealed that 1182 mRNAs, 157 transcripts with other functions, and 397 lncRNAs, including HOTAIR, TTC39A-AS1, DSCAM-AS1, and GATA3-AS1, are overexpressed in luminal subtype B. On the other hand, 743 mRNAs, 228 lncRNAs, and 61 transcripts with other functions were downregulated in luminal subtype B patients (Figure 3A). To evaluate the expression profile of luminal subtypes A and B, the results of differential expression analysis were used to generate a heatmap using nonhierarchical clustering with the top 40 lncRNAs differentially expressed in luminal tumors (Appendix A). This analysis showed that the lncRNAs DSCAM-AS1, GATA3-AS1, and TTC39A-AS1 were overexpressed in both luminal subtypes. In contrast, LINC01016 was specifically expressed in luminal A-subtype cell lines, and VIPR1-AS1 was identified as an lncRNA expressed in luminal B-subtype cell lines (Figure 3B,C).

To demonstrate the differences between subtypes in the expression levels of lncRNAs revealed using differential expression analysis, the expression levels of GATA3-AS1, LINC01016, TTC39A-AS1, and DSCAM-AS1 were then normalized in transcripts per million (TPM) format and compared between the four subtypes; the results showed that the expression of these lncRNAs was greater in luminal subtypes than in basal and HER2-enriched subtypes, as was observed for GATA3-AS1 and DSCAM-AS1 (*p* < 0.01, Wilcoxon test, Figure 4). Taken together, these results suggest the existence of a specific lncRNA profile for luminal BC cell lines; specifically, GATA3-AS1, DSCAM-AS1, and particularly luminal B-specific lncRNAs, such as VIPR1-AS1, could be related to the clinical features associated with the development of luminal breast cancer.

### 2.2. Transcriptomic Profiling of lncRNAs in Patients with Locally Advanced Breast Cancer

Transcriptome analysis of breast cancer cell lines revealed that the expression profiles of lncRNAs, such as GATA3-AS1 and DSCAM-AS1, are luminal specific [41]. Thus, to determine whether these expression profiles are consistent with the transcriptome in tumor samples, a discovery phase of transcriptome analysis was performed with 320 patients from the GEO datasets (GSE123845 [42] and GSE163882 [43]), which represent two independent cohorts of patients with LABC, to identify the lncRNAs differentially expressed in each breast cancer molecular subtype, particularly in the luminal B subtype. The results of this analysis were used to construct a heatmap using unsupervised hierarchical clustering, which revealed that luminal tumors have a characteristic lncRNA expression profile, and some differentially expressed lncRNAs include GATA3-AS1, LINC01016, and DSCAM-AS1, which were previously identified in luminal breast cancer cell lines (Figure 3), and an additional lncRNA, MAPT-IT1, which was found to be overexpressed in luminal breast cancer patients (Figure 5). Similarly, this analysis revealed the lncRNA expression profile of basal-like tumors and an expression profile that was independent of the molecular subtype (Figure 5).

Since these lncRNAs are expressed in luminal B patients, we aimed to determine whether they can be used as a signature to distinguish between NAC-sensitive and NAC-resistant luminal tumors. Thus, RNA-Seq data for 150 luminal tumors were filtered from the GSE123845 and GSE163882 cohorts, and their lncRNA profiles were subsequently analyzed. In both cohorts, the analysis revealed that there was homogeneity in the transcriptional profiles of pCR after NAC and RD after NAC patients, as indicated by a high correlation between all transcriptional profiles (Pearson correlation >0.7 in all patients). This indicates that the two groups of tumors maintain similar gene expression profiles, regardless of their response to NAC treatment.

Subsequently, the expression levels of the luminal subtype-specific lncRNAs LINC01016, GATA3-AS1, and MAPT-IT1 (Figure 6) were evaluated in the GSE123845 and GSE163882 cohorts, as well as in two independent validation cohorts including 4307 American patients (TCGA) [44] and Scandinavian patients (The Sweden Cancerome Analysis Network-Breast, SCAN-B) [45]. These results show that LINC01016 expression was significantly greater in luminal A and B subtypes (*p* < 0.05, Wilcoxon test), but there were no significant differences in LINC01016 expression between luminal A tumors and luminal B tumors (Figure 6B). Moreover, MAPT-IT1 expression levels were significantly greater in both luminal subtypes than in basal and Her2-enriched tumors (*p* < 0.01, Wilcoxon test), although the difference between the luminal A and B subtypes was not statistically significant (Figure 6C). Finally, GATA3-AS1 expression levels were significantly greater in luminal B patients than in patients with all other subtypes, including luminal A patients (*p* = 0.02, Wilcoxon test, Figure 6A). This finding suggests that even if transcriptome profiles are similar, lncRNAs can filter subtypes, and hence the evaluation of response and eventually survival in this specific subtype. Furthermore, according to the scientific literature on breast cancer tumors, these three lncRNAs are related to cell proliferation, which is a biological process related to resistance to treatment [37,39,46]. Thus, the expression of the luminal subtype-specific lncRNAs LINC01016, MAPT-IT1, and GATA3-AS1 could be related to NAC treatment response in luminal tumors.

To validate whether MAPT-IT1 and GATA3-AS1 are associated with resistance (RD) to NAC in luminal tumors, we performed a differential gene expression analysis comparing RD patients and pCR patients from GSE123845 and GSE163882 cohorts (Figure 7A,B), and their overexpression was validated in an independent cohort of 20 Hispanic luminal breast cancer patients (GSE270967, Figure 7C,D). In addition, MAPT-IT1 and GATA3-AS1 were found to be significant predictors of treatment-resistant disease according to an age-adjusted multivariate logistic regression model (*p* < 0.01 for both). Taken together, these results show that lncRNAs, particularly MAPT-IT1 and GATA3-AS1, are specifically expressed in luminal breast cancer tumors, and that their expression could be related to NAC response in RD patients.

To determine the potential functional roles in which these lncRNAs could be involved in the NAC tumor response, we performed a GSEA with the differential gene expression analysis results from GSE123845 and GSE163882 cohorts. This analysis showed that the differentially expressed mRNAs and lncRNAs in luminal tumors that presented RD to neoadjuvant chemotherapy are involved in biological processes related to neuronal processes, such as glutamaergic and serotonergic synapses, as well as pathways related to hormone response, such as the estrogen signaling pathway (Appendix A). Moreover, a set of metabolic processes were found to be differentially enriched, such as those related to linoleic acid metabolism, which have been previously associated with neoadjuvant chemotherapy response in breast cancer. Additionally, we performed an lncRNA correlation analysis with the lncGSEA tool in order to determine the gene sets correlated with GATA3-AS1, FLJ12825, FAM222A-AS1, and MAPT-IT1 lncRNA signature in breast cancer cohort from TCGA project. We identified 1368 gene sets related to cancer differentially enriched in breast tumors (Appendix A), from which 27 are gene sets related to breast cancer features (Appendix A) and 12 are related to hallmarks of cancer (Appendix A). Altogether, these results suggest the association of these NAC resistance-related lncRNAs to the development of resistance to treatment in breast cancer tumor cells.

In particular, from the NAC resistance-related lncRNA signature, GATA3-AS1 is of particular interest, because it is an lncRNA whose genomic location is adjacent to that of GATA3 [46], a transcription factor related to luminal breast cancer tumor development, since GATA3 regulates cell proliferation and differentiation in mammary cells [47,48]. Previous reports in the scientific literature associated GATA3-AS1 expression detection using RT–qPCR in biopsies from frozen tissue of luminal breast cancer NAC-resistant patients [27], but there is no information about GATA3-AS1 detection in paraffin-embedded tissue, which is the most common sample type for laboratory tests in oncology.

### 2.3. Experimental Validation Analysis of GATA3-AS1 in Patients with Locally Advanced Breast Cancer

In our previous work, GATA3-AS1 was established as an independent predictor of NAC response in luminal B breast cancer patients (odds ratio, 37.49; 95% CI, 6.74–208.42; *p* = 0.001). Thus, we next validated the clinical utility of GATA3-AS1 as a predictive biomarker of NAC response in a retrospective cohort using RT–qPCR and RNA-ISH using RNAScopeTM technology [49]. Since GATA3-AS1 expression correlates with GATA3 expression in breast tumors [27], we analyzed GATA3-AS1 and GATA3 expression levels by RT–qPCR of cDNA from paraffin-embedded tissue from 49 pre-NAC Hispanic LABC luminal subtype B tumors using two different housekeeping genes (RPLP0, Figure 8A,B. RPS28, Figure 8C,D). This analysis revealed that the relative expression levels of GATA3-AS1 to RPLP0 in the 9 patients who achieved pCR were significantly lower than those in the 40 patients who achieved RD (*p* = 0.029) (Figure 8A). In contrast, Figure 8B shows the relative expression of GATA3 in the pCR group compared to that in the RD group. In this case, the difference between the two groups was not statistically significant (*p* = 0.29). Additionally, analysis of the relative expression levels of GATA3-AS1 (Figure 8C) and GATA3 (Figure 8D) to RPS28 showed that GATA3-AS1 (*p* = 0.036), but not GATA3 (*p* = 0.196), was overexpressed in RD patients.

Meanwhile, the analysis of an ROC Curve showed that GATA3-AS1 detection from FFPE samples using RT-qPCR has associated a high specificity and sensitivity for RD after NAC prediction (>80% and >90%, respectively. Figure 9A). Taken together, our results indicate that GATA3-AS1 is a luminal B-specific lncRNA that is associated with pCR and that its overexpression has predictive value for NAC resistance in luminal B breast cancer patients, but its adjacent gene, GATA3, is not a predictive biomarker in this group of patients.

Moreover, since pCR is a surrogate prognostic indicator and could be related to the clinical outcome of breast cancer patients, we next analyzed the association of GATA3-AS1 expression in a retrospective cohort of patients. Our results reveal significant differences between the clinical characteristics of pCR patients and RD patients (Table 1), since the median event-free survival of the complete cohort was 3.4 years (range 1.9–6.8), while the median event-free survival of the nonresponder group with RD was 2.7 years (range 1.8–5.3, Figure 9B); the median event-free survival of the responder patients with pCR was significantly longer (7.0 years, range 5.6–8.6, *p* = 0.04, Cox model). Moreover, we ran an age-adjusted multivariate regression model for the predictive value of GATA3-AS1 as a predictor variable for RD. Our results indicate that GATA3-AS1 is an independent predictor for RD, with an OR of 1.48 (95% CI 1.12–2.25, *p* = 0.024). However, there were no significant differences in overall survival (OS) between patients who overexpressed GATA3-AS1 (Appendix A, above the median, red) and those who did not overexpress this lncRNA (Appendix A, below the median, blue). In conclusion, GATA3-AS1 has associated predictive, but no OS prognostic value in Hispanic luminal B LABC patients.

Taken together, our results demonstrate that GATA3-AS1 could be clinically applicable as a predictive biomarker for NAC efficacy in luminal B breast cancer patients according to an RT–qPCR analysis of paraffin-embedded tissues. However, it is common in oncology laboratory tests to identify biomarkers in breast tumor samples using histological techniques, such as immunohistochemistry, for the detection of detect biomarkers such as Ki67 [50] and estrogen receptor [6] and in situ hybridization for the detection of HER2 amplification [51]. Thus, we next aimed to evaluate whether GATA3-AS1 could be detected in histological slides of breast tumor samples using RNA-ISH in a retrospective cohort of Hispanic luminal B breast cancer patients. We analyzed the expression of the lncRNA GATA3-AS1 in whole sections of FFPE breast biopsies of locally advanced, invasive luminal B breast carcinoma after NAC, but no other prior treatment. The samples from patients with RD were given a score of 2 (Figure 10, right panel) on the RNA-ISH signal scale, whereas those patients with pCR were given a score of 0 (Figure 10, left panel). These results indicate that the expression of GATA3-AS1 determined using RNA-ISH was greater in RD patients than in pCR patients. According to these findings, we concluded that GATA3-AS1 detected using RNA-ISH can be used as a biomarker for predicting pathologic complete response in patients with locally advanced luminal B breast carcinoma, and that RNA-ISH is an adequate technique for determining its expression.

## 3. Discussion

Among the breast cancer molecular subtypes, luminal breast cancer is the most commonly diagnosed subtype worldwide [52], and is associated with better clinical outcomes among breast cancer patients [53]. However, locally advanced luminal breast cancer patients, particularly luminal B patients, have a lower pCR rate after NAC [54] due to their molecular characteristics, since these patients have heterogeneous tumors with variable clinical outcomes [55]. Furthermore, luminal B tumors are the most frequently diagnosed breast cancers and remain undefined and without the appropriate treatment [41,56], so new molecular biomarkers need to be developed to differentiate luminal B patients who will benefit from NAC from those unlikely to respond to treatment, leading to an improvement in clinical decision-making and preventing related issues such as overtreatment.

Moreover, lncRNAs have emerged as potential candidates for the development of novel predictive biomarkers for the response to neoadjuvant chemotherapy since they exhibit tissue- and disease-specific expression [29]. Furthermore, there are lncRNAs with expression specificity among breast cancer molecular subtypes, such as LINC02188 in basal tumors, LINC00511 in HER2-enriched tumors, and GATA3-AS1 in luminal tumors, as reported by Xia and collaborators [57]. Furthermore, there is scientific evidence of the luminal subtype-specific expression pattern in luminal breast cancer tumors, as is the case for DSCAM-AS1, which is related to estrogen receptor function in breast cancer luminal cells [30], and GATA3-AS1, which has also been related to the predictive value of neoadjuvant chemotherapy [27]. In our study, we performed a three-phase analysis: a pre-clinical phase, a discovery phase, and a validation phase. The pre-clinical phase was integrated using a transcriptome BC cell line analysis, from which we identified an expression profile of lncRNAs that are expressed in luminal BC cell lines, such as GATA3-AS1 and LINC01016, and we also identified lncRNAs with expression specificity in luminal A tumors, such as LINC01016 and VIPR1-AS1, which are expressed only in luminal B cell lines and are novel lncRNAs related to the luminal phenotype. Then, we analyzed the transcriptomes of 320 tumor samples from the GEO dataset cohorts in the discovery phase, in which we identified a luminal subtype-specific lncRNA expression profile integrated using the lncRNAs LINC01016, GATA3-AS1, MAPT-IT1, and DSCAM-AS1, which has been previously described in the literature as a luminal subtype-related lncRNA [30] that were also validated in TCGA and SCAN-B cohorts (Figure 6). We also performed a validation analysis in a third independent cohort from Hispanic luminal B breast cancer patients (Figure 7B), in which we corroborated the overexpression of MAPT-IT and GATA3-AS1 in NAC-resistance breast cancer patients, which are lncRNAs that have been previously identified in fresh frozen tumor samples from breast cancer Hispanic luminal B patients [27]. Due to the biological heterogeneity that characterizes breast cancer tumors [58], which is enhanced by the race and ethnicity intrinsic features of the patients [59], we identified discrepancies in the expression profile of the NAC-resistance related lncRNA signature in the Hispanic cohort (Figure 7B) compared to the Korean and US cohorts (Figure 7A). This ancestry-related heterogeneity has been reported before in other genomic and molecular analysis of breast tumors and their response to treatment [60,61]. Since lncRNAs are characterized by their expression specificity among human tissues, it is possible that lncRNA profiles are also related with race and ethnicity characteristics. Further research is necessary in order to elucidate the association of lncRNA expression profiles and ancestry-related heterogeneity.

From our results from the differential gene expression analysis in the Hispanic cohort, we identified that MAPT-IT and GATA3-AS1 are lncRNAs overexpressed in NAC-resistant breast tumors, regardless of the ancestry of the patients in the cohort. Particularly, the lncRNA GATA3-AS1 have been previously described as a NAC response-related lncRNA in Hispanic breast cancer patients [27]. Thus, we performed a further validation of GATA3-AS1 association with the response to neoadjuvant chemotherapy in two independent cohorts of Hispanic patients using two different reference genes (RPLP0 and RPS28) for gene quantification using RT–qPCR, and the results corroborated the robustness of GATA3-AS1 as a biomarker associated with a predictive value for RD after NAC treatment, since this lncRNA presents a high specificity (91.5%) and sensitivity (83.0%, Figure 9A). This result also suggests the association between GATA3-AS1 expression and RD after NAC presence, since this lncRNA is overexpressed in those patients with reported RD in their clinical follow-up. However, although OS analysis showed no prognostic value for GATA3-AS1, the patient cohort analyzed in this study showed that pCR patients, who do not have overexpressed GATA3-AS1, have longer event-free survival events compared with those who present RD. Thus, since pCR is a surrogate of prognosis, GATA3-AS1 could also be used as a support diagnostic tool in conjunction with pCR evaluation for the improvement of prognosis determination in luminal B LABC after NAC.

Since GATA3-AS1 is consistently expressed in luminal B patients according to different studies [27,57] and was found to be associated with the RD after NAC in this group of patients, we aimed to detect GATA3-AS1 expression using RNA-ISH technology; this approach involves the use of histological slides, and is similar to the approach used to detect estrogen receptor, Ki67, and HER2 expression in breast cancer laboratory tests [8,49,51]. The results indicate that GATA3-AS1 is a feasible biomarker for routine clinical analysis. The detection of GATA3-AS1 in a retrospective cohort of Hispanic luminal B patients supported the finding that this lncRNA is detectable in histological slides, and a high score for this lncRNA was related to RD after NAC in these patients. Furthermore, pragmatism of the new era is not waiting for survival, so surrogates as pCR are mostly used as the standard, but its accuracy could be an issue, so the use of biomarkers to identify patients who will not benefit from NAC treatment should be encouraged. Most trials evaluating the utility of new drugs use pCR as the primary outcome for its time and cost-effectiveness; thus, RNA-ISH could even shorten the gap in luminal B patients [58], representing a feasible option to be applied for breast cancer tests in clinical routine.

However, several follow-up experiments, including the analysis of GATA3-AS1 expression using RNA-ISH in an independent cohort of patients, are needed to validate the results obtained in this study. Additionally, in this study, we analyzed only lncRNAs whose expression was associated with luminal breast cancer subtypes, but further analysis is needed to identify the coding genes that are coexpressed with these lncRNAs. Gene ontology enrichment analysis and gene set enrichment analysis (GSEA) also need to be performed to assess related biological processes. Moreover, experiments are needed to evaluate a combined signature integrated with protein-coding genes and lncRNAs that could be used to distinguish between luminal A and luminal B tumors, which is an opportunity for the development of new diagnostic and prognostic biomarkers for breast cancer. Lastly, this study is only focused on analyzing the lncRNA differentially expressed in luminal B patients using RNA-Seq transcriptome analysis. Although bulk transcriptome has led to the identification of molecular biomarkers in breast cancer, such as the expression signature Oncotype Dx [22], this approach does not allow us to identify the source of the gene expression in the cellular heterogeneity that integrates de tumor microenvironment. This limitation is solved with single cell analysis [62] or spatial transcriptomics [63], since both methodologies allow for the characterization of the cellular heterogeneity in the breast tumor microenvironment. It has been shown that lncRNA’s location in the tumor microenvironment is related to clinical features [64]. Thus, the implementation of spatial transcriptomics in the study of luminal B tumors is needed in order to determine the association of the lncRNAs’ transcription location in the tumor microenvironment with treatment response in luminal B breast tumors.

## 4. Materials and Methods

### 4.1. Total RNA Extraction

Total RNA from each sample was extracted using the RNeasy FFPE Kit according to the manufacturer’s specifications (QIAGEN^®^, Venlo, The Netherlands), and was subsequently quantified using a spectrophotometer (Nanodrop, Thermo Fisher Scientific^®^, Waltham, MA, USA). RNA integrity was assessed using a bioanalyzer (Tapestation 2200, Agilent Technologies^®^, Santa Clara, CA, USA); only samples with an RNA integrity number (RIN) >2.0 were used for RT–qPCR. Thirty-six micrograms of total RNA was stored in an ultrafreezer at −80 °C to ensure its integrity until validation using RT–qPCR.

### 4.2. Real-Time PCR (RT–qPCR) Validation

After total RNA was extracted, cDNA was synthesized via reverse transcription using a reverse transcription system kit (Promega^®^, Madison, WI, USA). Finally, RT-qPCR was performed in technical triplicates using SYBR GreenTM real-time master mixes (Thermo Fisher Scientific^®^). As a control for expression analysis, values were normalized to those of constitutively expressed coding RNA (RPLP0). Finally, expression was quantified using the delta Cq (∆Cq) method.

### 4.3. RNA ISH

GATA3-AS1 RNA in situ hybridization (ISH) was performed using RNAscope™ Technology (ACD, Carlsbad, CA, USA) following the manufacturer’s instructions (Advanced Cell Diagnostics, Inc., Hayward, CA, USA) on formalin-fixed paraffin-embedded (FFPE) tissue sections (5 μm) from patients with locally advanced luminal B (HER2-, HER2+) breast cancer treated at the National Cancer Institute. The patients exhibited either a complete pathologic response or no complete pathologic response to neoadjuvant chemotherapy treatment. The samples were pretreated with xylene and ethanol for deparaffinization and dehydration. Subsequently, they were submerged in target retrieval boiling solution (98–102 °C) for 15 min. Protease treatment was then applied to permeabilize and allow for the penetration of predesigned probes for GATA3-AS1 (Cat#: 1277751-C1, Advanced Cell Diagnostics, Inc., Hayward, CA, USA). The samples were incubated at 40 °C in a HybEZ™ oven (Hybridization system HyBEZ II ACD. Cod.S.A.T. 41116133, Newark, CA, USA) for 2 h. Next, a six-step amplification process was carried out, and the samples were incubated at 40 °C for 15 to 30 min to amplify target-specific signals. Chromogenic detection was performed using DAB, followed by counterstaining with hematoxylin. Signal evaluation was conducted under a bright-field microscope at 20–40× magnification.

### 4.4. RNA Sequencing for Independent Validation Cohort

Twenty RNA samples, obtained from biopsies of female patients diagnosed with locally advanced mammary adenocarcinoma (stage IIB to IIIC) belonging to the Mexican National Cancer Institute population (Hispanic patients) and who were candidates for the administration of neoadjuvant chemotherapy were sequenced using RNA-Seq. To ensure good quality of the samples for sequencing, Nanodrop, Qubit 2.0 (Life Technologies, Carlsbad, CA, USA), and the Agilent 2100 Bioanalyzer (Agilent Technologies) were used to detect the purity, concentration, and integrity of RNA samples, respectively. All samples had RNA integrity numbers >8.0. A total of 1 μg RNA was used to generate sequencing libraries using the TruSeq Stranded mRNA library prep kit from Illumina, Inc. (San Diego, CA, USA), according to the manufacturer’s instructions. After construction of the libraries, their concentrations and insert sizes (approximately 260 bp) were detected using Qubit 2.0 and the Agilent 2100 Bioanalyzer, respectively. The library was then sequenced using an Illumina HiSeq 2500 sequencer with paired-end 2 × 125 cycles using Illumina TruSeq version 4 sequencing using synthesis (SBS) chemistry and by following the manufacturer’s instructions. The depth of sequencing was >50 million reads. RNA-Seq from Hispanic Cohort data are available from National Center for Biotechnology Information GEO (https://www.ncbi.nlm.nih.gov/geo, accession number GSE270967. Accessed on 15 July 2024).

### 4.5. Bioinformatic Analysis

Two independent cohorts were included in the analysis: a Korean cohort [42], and a US cohort [43]. RNA-Seq data from a total of 320 malignant breast tumors were downloaded from the National Center for Biotechnology Information (NCBI) public library (GEO datasets, IDs GSE123845, and GSE163882), which included complete clinical data (age, tumor grade, menopausal status, response to NAC according to pCR or RD presence, lymphocyte infiltration, and tumor immunophenotype). Sequencing files in .fastq format were processed using a preprocessing protocol for paired transcriptomic data (quality control, normalization, filtering, and annotation) using fastq-dump tools version 2.9.1, fastQC version 0.11.9 [65], multiQC version 1.9 [66], and Trimmomatic version 0.32 [67]. After filtering, read alignment of the sequencing data against the human reference genome (ENSEMBL, GRCh38.p13) was performed [68]. The use of any pipeline to analyze RNA-Seq data introduces a percentage error in the estimation of transcript abundance. Therefore, two different pipelines with different features were used for data analysis: an aligner, STAR (version 2.7.7a) [69], and a pseudoaligner, Salmon (version 1.4.0) [70]. Once the sequencing data were aligned, the matrix of transcript counts was extracted and imported into the data analysis software RStudio (version 4.0.3) [71]. Differential gene expression analysis was performed using DESeq2 (version 1.30.1) [72]. Transcripts with adjusted *p* values < 0.01 and log2-fold change values less than −2.0 or greater than 2.0 were considered to be differentially expressed. Only transcripts identified using both STAR and Salmon were considered differentially expressed. Finally, we performed a gene set enrichment analysis (GSEA) with fgsea (v1.30.0) [73] and lncGSEA (v0.1.0) [74].

Prior to differential expression analysis, a general quality control analysis was performed, including principal component analysis (PCA), to identify the variables that contributed most to the variability of the data. Based on the results of the quality control analysis, tumor purity was included as a covariate in the model design (~tumor purity + pCR). Finally, we searched for lncRNAs differentially expressed in the different molecular subtypes of BC, as well as in NAC-resistant tumors of patients with luminal B LABC.

### 4.6. Statistical Analysis

Demographic and clinical variables were evaluated using univariate and bivariate analyses. The Shapiro–Wilk test was used to assess the distribution of the data. Bivariate analysis of quantitative variables between patients who achieved a pCR and those who presented RD was performed using Student’s *t* test or the Mann–Whitney U test, depending on their distribution. A Pearson correlation was calculated between each of the tumor transcriptional profiles to determine the correlation between phenotype and lncRNA expression. A chi-squared test was used to evaluate qualitative variables. Logistic regression models were constructed to determine the effect of candidate lncRNAs on patients’ pCR or RD, adjusting for other measured clinical variables. In addition, a machine learning model based on the random forest algorithm was constructed using the caret [75] and randomForest [76] R packages. The model was trained by generating a subset of data corresponding to 70% of the cohort, and was subsequently tested on the remaining 30%. In addition, bootstrapping was used in 25 iterations to simulate values other than those contained in the internal cohort and to validate the model, allowing for real estimates of its performance in external cohorts. All analyses were performed using RStudio version 4.2.1. [71]. *p* values < 0.05 were considered to indicate statistical significance. To determine the discriminative ability of GATA3-AS1 as a biomarker for predicting response to NAC, the C-statistic of the predictive model was calculated, and an area under the curve (AUC) plot was constructed.

## 5. Conclusions

In conclusion, our study revealed the existence of an lncRNA signature specific for breast cancer tumors, and one of those lncRNAs, GATA3-AS1, is a luminal B-associated lncRNA that can be used to predict the response to neoadjuvant chemotherapy, and could be included in routine clinical practice as a predictive biomarker because of its ability to be detected using RNA ISH.

## Figures and Tables

**Figure 1 ijms-25-08077-f001:**
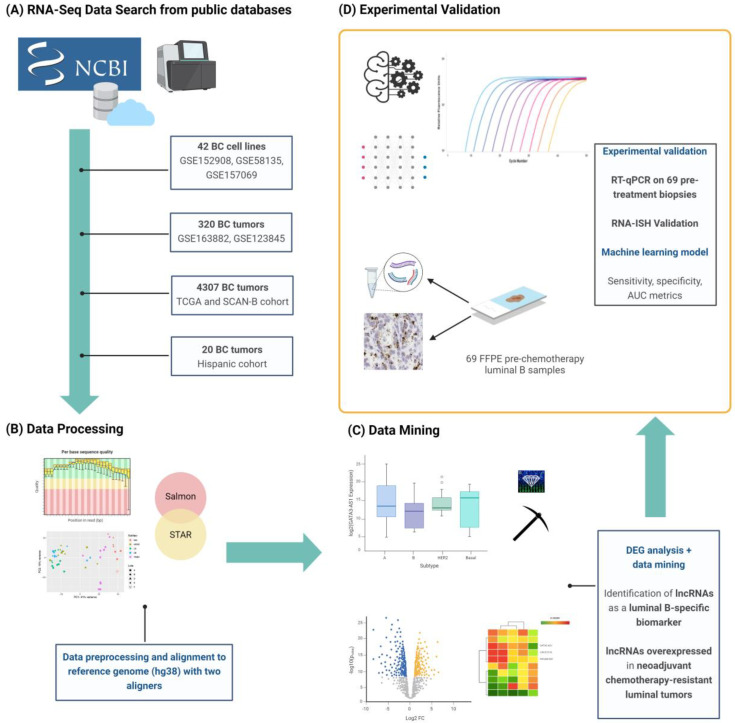
**Bioinformatic analysis and experimental validation workflow.** (**A**) Publicly available RNA-Seq data from 42 breast cancer cell lines, where 320 breast cancer tumors were downloaded from GEO datasets. Two independent cohorts from The Cancer Genome Atlas (TCGA) and Scandinavian patients (The Sweden Cancerome Analysis Network-Breast, SCAN-B) projects (n = 4307) were included for analysis, and a third independent cohort was from Hispanic patients (GSE270967, n = 20). (**B**) Following preprocessing and quality control, counts were aligned and differential expression analysis was performed. The results from the bioinformatic analysis coupled with (**C**) data mining revealed that lncRNAs are luminal subtype-specific and are overexpressed in chemotherapy-resistant tumors of this subtype. GATA3-AS1 was identified as a luminal B-specific lncRNA, and (**D**) its expression was validated using qPCR and RNA-ISH.

**Figure 2 ijms-25-08077-f002:**
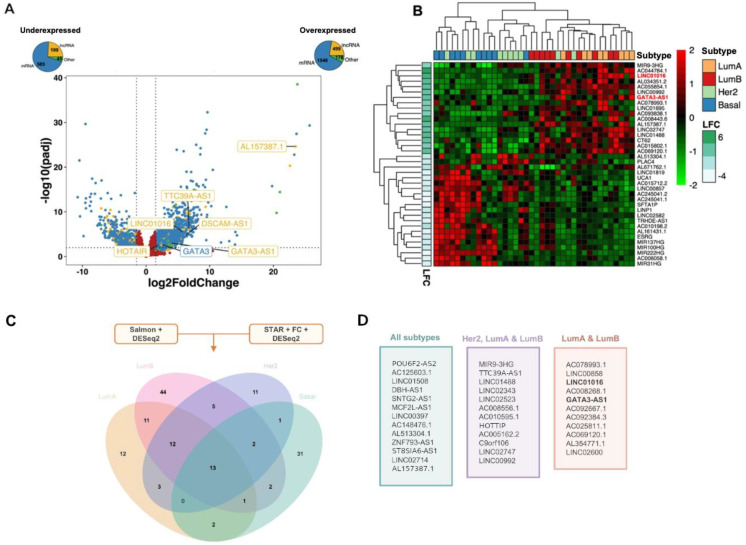
**Identification of deregulated lncRNAs in breast cancer models.** (**A**) Volcano plot showing the 2874 differentially expressed genes in BC cell lines corresponding to the four molecular subtypes of breast cancer (basal, Her2-enriched, luminal A, and luminal B), which shows a group of transcripts differentially expressed in the upper left and right quadrants. At the top of the graph, pie charts indicate the biotypes of differentially expressed genes and the total number of genes in each category: 1546 mRNAs, 499 lncRNAs, and 176 transcripts with other functions were upregulated, while 585 mRNAs, 199 lncRNAs, and 45 transcripts with other functions were downregulated. In graph, differentially expressed mRNAs are shown in blue dots, and lncRNAS in yellow dots, while genes with not changes are shown in red. Horizontal dotted line indicates the p adjusted value threshold (0.05) and vertical dotted lines indicate log2 Fold Change cutoff values (−1.5 and 1.5). (**B**) Heatmap showing the top 40 differentially expressed lncRNAs in BC cell lines of the four molecular subtypes of breast cancer, including two lncRNAs with specific expression for luminal subtypes. The lncRNAs that are overexpressed are presented in red, while those lncRNAs that are underexpressed are presented in green. The color scale is normalized and plotted as z scores, which were derived from the transcripts per million (TPM) of each lncRNA (rows) per tumor (columns). The left column of the heatmap represents the relative their log2-fold change (LFC) of each transcript in malignant cell lines compared to nonmalignant cell lines. LncRNAs LINC01016 and GATA3-AS1 are highlighted in red. (**C**) Venn diagram showing the 150 differentially expressed lncRNAs identified in both analyses (Salmon + DESeq2 and STAR + fc + DESeq2) distributed among the breast cancer molecular subtypes. The intersections highlight the number of lncRNAs differentially expressed and shared between groups, which are listed in (**D**) colored boxes. The lncRNAs differentially expressed among all breast cancer subtypes (green) and in the HER2-enriched and luminal subtype (purple) and luminal subtype only (orange) are highlighted. In luminal subtype lncRNAs (orange), lncRNAs LINC01016 and GATA3-AS1 are highlighted in black.

**Figure 3 ijms-25-08077-f003:**
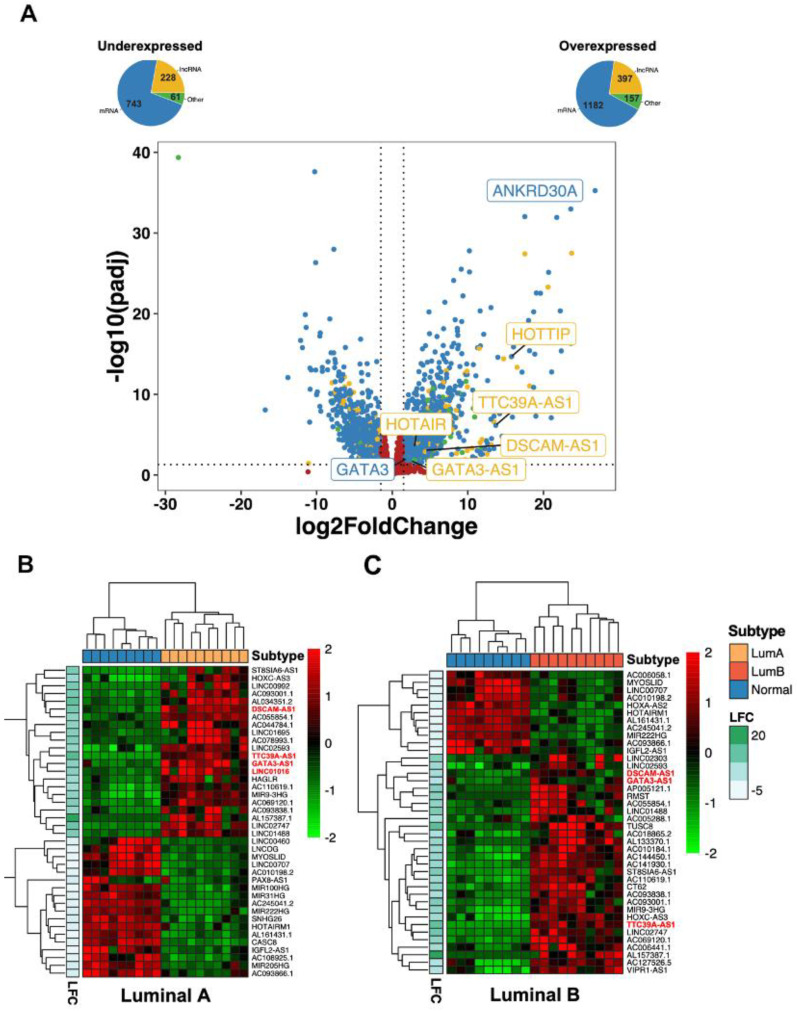
**Identification of deregulated lncRNAs in luminal breast cancer cell lines.** (**A**) Volcano plot showing the results of differential gene expression analysis between nonmalignant and luminal B breast cancer patients, showing a group of differentially expressed lncRNAs, including GATA3-AS1, DSCAM-AS1, TTC39A-AS1, and HOTTIP, which have been previously implicated in breast cancer pathogenesis. The pie charts show that 1182 mRNAs, 397 lncRNAs, and 157 transcripts with other functions were upregulated, while 743 mRNAs, 228 lncRNAs, and 61 transcripts with other functions were downregulated. In graph, differentially expressed mRNAs are shown in blue dots, and lncRNAS in yellow dots, while genes with not changes are shown in red. Horizontal dotted line indicates the p adjusted value threshold (0.05) and vertical dotted lines indicate log2 Fold Change cutoff values (−1.5 and 1.5). (**B**,**C**) Heatmaps showing the 40 lncRNAs with the greatest variance in malignant luminal A (**B**) and luminal B (**C**) breast cancer cell lines compared to nonmalignant cell lines. Three lncRNAs, GATA3-AS1, DSCAM-AS1, and TTC39A-AS1, exhibit luminal subtype-specific expression. LINC01016 exhibited luminal A-specific expression (highlighted in red).

**Figure 4 ijms-25-08077-f004:**
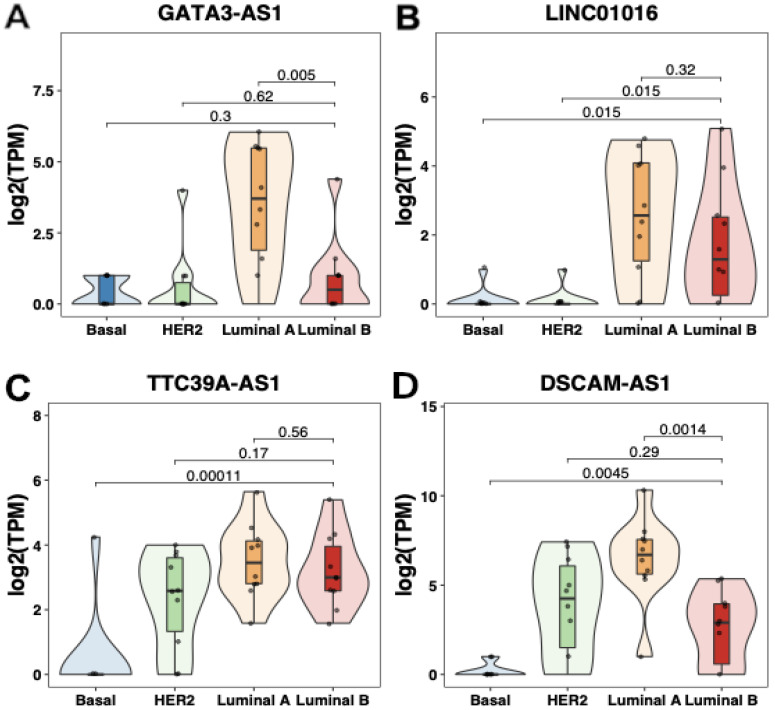
**lncRNAs with specific expression in luminal cell lines.** (**A**) Expression of GATA3-AS1 in the different molecular subtypes of breast cancer. When comparing luminal B tumors with basal and Her2-enriched tumors, the difference in expression levels was not statistically significant. GATA3-AS1 expression was greater in luminal A tumors than in the other subtypes (*p* < 0.01, Wilcoxon). (**B**) LINC01016 expression in the different molecular subtypes of breast cancer. LINC01016 expression was greater in the luminal B subtype than in the basal and Her2-enriched subtypes (*p* < 0.05, Wilcoxon); the difference in expression levels between the luminal A and B subtypes was not statistically significant. (**C**) TTC39A-AS1 expression in the different molecular subtypes of breast cancer. TTC39A-AS1 expression levels were significantly greater in the luminal B subtype than in the basal subtype (*p* < 0.01, Wilcoxon); there were no significant differences in expression levels between the Her2-enriched, luminal A, and luminal B subtypes. (**D**) DSCAM-AS1 expression in different breast cancer subtypes. DSCAM-AS1 expression levels were significantly greater in the luminal B subtype than in the basal subtype (*p* < 0.01, Wilcoxon); the difference in expression levels between the Her2-enriched and luminal B subtypes was not statistically significant. DSCAM-AS1 expression levels were significantly greater in the luminal A subtype than in the luminal B subtype (*p* < 0.01, Wilcoxon).

**Figure 5 ijms-25-08077-f005:**
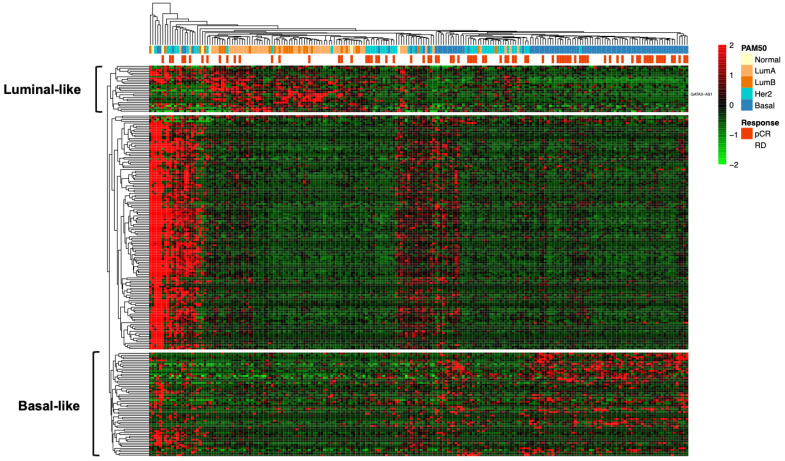
**Identification of a luminal subtype-specific lncRNA expression profile in clinical samples**. Unsupervised hierarchical clustering revealed three groups of all deregulated lncRNAs in breast tumors: a “luminal-like” cluster, a “basal-like” cluster, and a mixed cluster from the GSE123845 and GSE163882 cohorts. GATA3-AS1, LINC01016, and DSCAM-AS1 are overexpressed in tumors in the “luminal-like” cluster. The PAM50 subtype and response to treatment with chemotherapy are displayed as rows above the heatmap.

**Figure 6 ijms-25-08077-f006:**
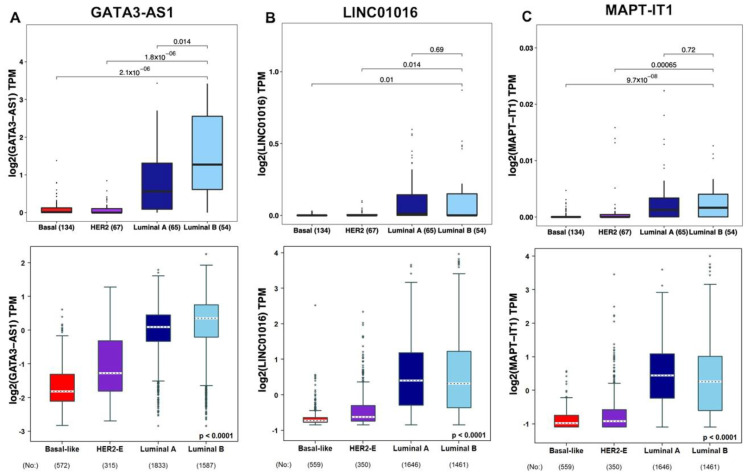
**Subtype-specific lncRNA expression levels in clinical samples.** Box plots showing the expression levels of GATA3-AS1, LINC01016, and MAPT-IT1 in the GSE123845 and GSE163882 cohorts (n = 320, upper panels) and in the SCAN-B and TCGA cohorts (n = 4307, lower panels). (**A**) GATA3-AS1 expression levels were significantly greater in the luminal B subtype than in all other subtypes. (**B**) LINC01016 expression levels were significantly greater in both luminal subtypes, but no significant differences were detected between luminal A and B tumors. (**C**) MAPT-IT1 expression levels were significantly greater in both luminal subtypes, but no significant differences were observed between luminal A and B tumors. Wilcoxon’s rank sum test was used in all cases to assess differences in medians. Each black dot represents the expression value of the lncRNA for an individual patient.

**Figure 7 ijms-25-08077-f007:**
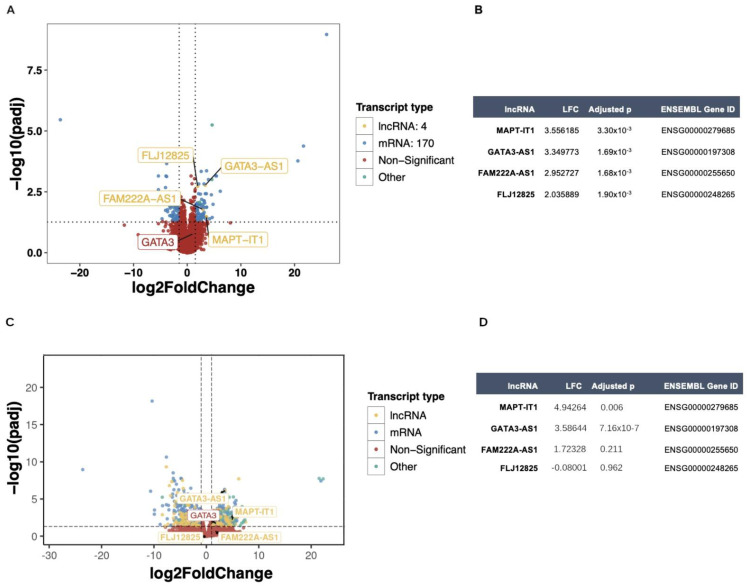
**Identification of dysregulated lncRNAs in NAC-resistant luminal B tumors.** (**A**) Volcano plot showing the results of DEA between responder and nonresponder patients. Four lncRNAs were overexpressed in NAC-resistant tumors: GATA3-AS1, FLJ12825, FAM222A-AS1, and MAPT-IT1 from GSE123845 and GSE163882 cohorts. The GATA3 gene was not significantly deregulated in this group of tumors (red dot). (**B**) Table of the four lncRNAs overexpressed in NAC-resistant disease: MAPT-IT1, GATA3-AS1, FAM222A-AS1, and FLJ12825, with their log2-fold change (LFC) values, associated adjusted *p* values, and ENSEMBL gene IDs from GSE123845 and GSE163882 cohorts. (**C**) Volcano plot showing the results of DEA between responder and nonresponder Hispanic breast cancer patients. Four lncRNAs were overexpressed in NAC-resistant tumors: MAPT-IT1, GATA3-AS1, FAM222A-AS1, and FLJ12825. The GATA3 gene was not significantly deregulated in this group of tumors (red dot) from an independent cohort of Hispanic luminal B breast cancer patients (n = 20). (**D**) Table of the corroboration of three lncRNAs overexpressed in NAC-resistant disease: MAPT-IT1, GATA3-AS1, FAM222A-AS1, and FLJ12825, with their log2-fold change (LFC) values, associated adjusted *p* values, and ENSEMBL gene IDs from an independent cohort of Hispanic luminal B breast cancer patients (n = 20). In A and C graphs, differentially expressed mRNAs are shown in blue dots, and lncRNAS in yellow dots, while genes with not changes are shown in red. Horizontal dotted line indicates the p adjusted value threshold (0.05) and vertical dotted lines indicate log2 Fold Change cutoff values (−1.5 and 1.5).

**Figure 8 ijms-25-08077-f008:**
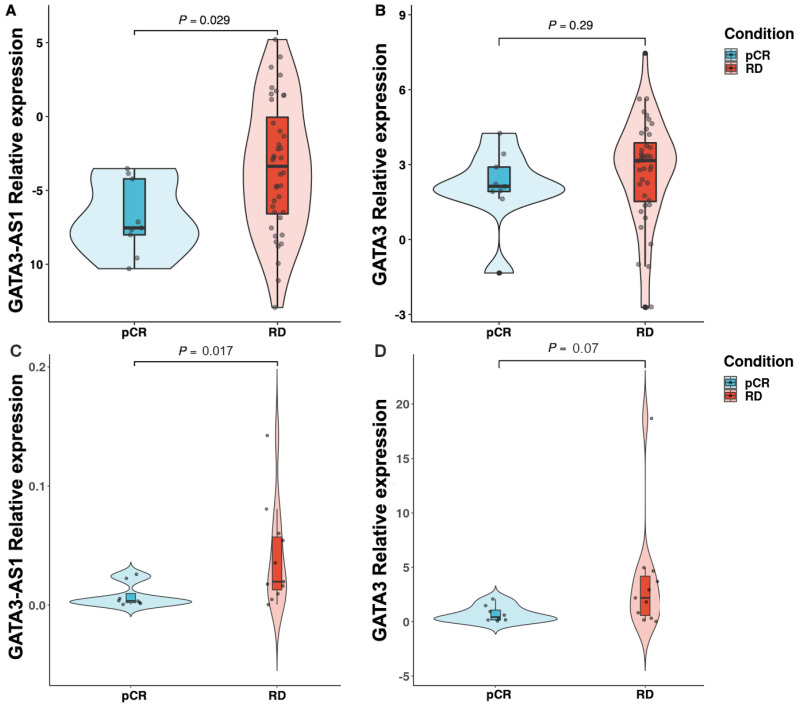
**GATA3-AS1 is overexpressed in neoadjuvant chemotherapy-resistant luminal B tumors.** (**A**) RT–qPCR analysis of GATA3-AS1 expression in FFPE tumor tissue revealed that its expression was significantly greater in RD patients (n = 40) than in pCR patients (n = 9) (*p* = 0.029, Wilcoxon rank sum test). (**B**) The expression levels of the protein-coding gene GATA3 were not significantly different between RD after NAC and pCR after NAC (*p* = 0.29, Wilcoxon rank sum test). The transcript expression levels were normalized to the expression levels of RPLP0, a constitutively expressed gene in breast tissue. (**C**) RT–qPCR analysis of the GATA3-AS1 expression levels in cDNA from FFPE tumor tissues from an independent validation cohort revealed that the expression of GATA3-AS1 was significantly greater in RD (n = 11) than in pCR patients (n = 9) (*p* = 0.017, Wilcoxon rank sum test). (**D**) Expression levels of the protein-coding gene GATA3 in an independent validation cohort were not significantly different between RD after NAC and pCR after NAC (*p* = 0.07, Wilcoxon rank sum test). The transcript expression levels were normalized to the expression levels of RPS28, a constitutively expressed gene in breast tissue. All samples were from Hispanic patients diagnosed with Luminal B breast cancer.

**Figure 9 ijms-25-08077-f009:**
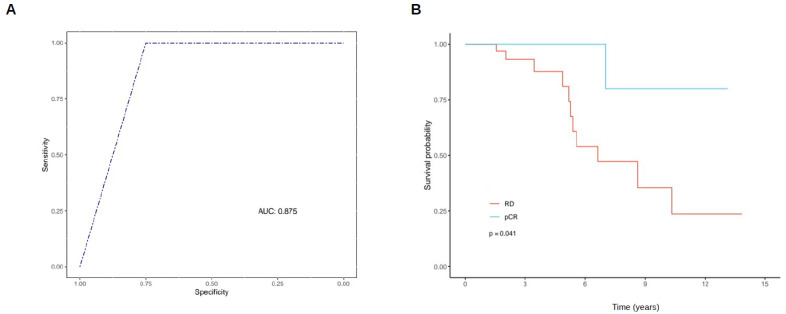
**GATA3-AS1 allows for the identification of luminal B LABC patients who will present RD after NAC treatment.** (**A**) ROC Curve construction to determine sensitivity (84.1%) and specificity (91.5%) of GATA3-AS1 detection using RT-qPCR to predict RD after NAC in luminal B LABC patients (n = 49, cutoff = 0.027). The area under the curve (AUC) of this ROC curve is 0.875. The x-axis presents the proportion of true negative cases (specificity), and the y-axis presents the true positive cases (sensitivity). (**B**) Kaplan–Meier curve showing the event-free survival for luminal B LABC patients after NAC treatment (n = 49, *p* = 0.041). All samples were from Hispanic patients diagnosed with Luminal B breast cancer.

**Figure 10 ijms-25-08077-f010:**
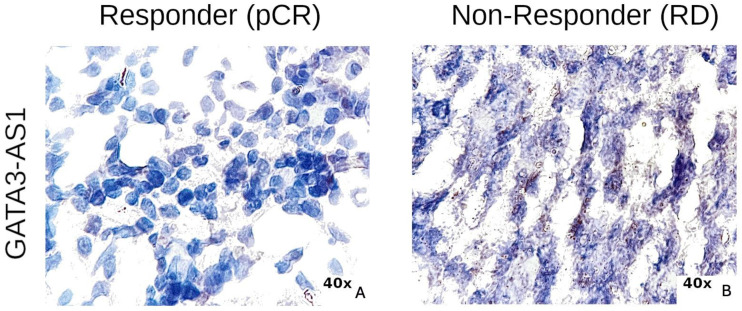
**The lncRNA GATA3-AS1 is detectable using RNA-ISH in locally advanced luminal B breast cancer**. Positive signals for GATA3-AS1 (brown dots) in (**A**) pCR patients and (**B**) RD patients. A total of 137 histological slides from patients with locally advanced luminal B breast carcinoma were analyzed (pCR, n = 7; RD n = 7). The assay results were visualized under a bright-field microscope, and semiquantitative scoring guidelines were used to evaluate the staining results. The signals observed as brown dots in the nuclei of epithelial neoplastic cells with blue contrast were counted. A total of 108 (78.83%) patients had 4 to 10 signals per 10 cells (score 2) at 40×/0.65 amplification. In the pCR group, the remaining 29 (21.16%) patients had less than 1 per 10 cells (score 0) at 40×/0.65 amplification. The number of dots correlates to the number of RNA copy numbers. (**A**) Representative sample for the pCR group; (**B**) representative sample for RD group. NAC: neoadjuvant chemotherapy; pCR: pathologic complete response. All samples were from Hispanic patients diagnosed with Luminal B breast cancer.

**Table 1 ijms-25-08077-t001:** Clinical characteristics of the luminal B LABC Patients in GATA3-AS1 expression from FFPE analysis. †: Student’s *t*-test, ƒ: Fisher’s exact test, §: Mann–Whitney’s U.

Clinical Characteristic	Resistant Disease (RD)	Pathological Complete Response (pCR)	*p* Value
Mean age, years (SD)	47 (9.9)	44 (10.3)	0.38 †
Menopause, n (%)			0.86 ƒ
Premenopause	24 (60%)	4 (44.4%)	
Postmenopause	16 (40%)	5 (55.6%)	
Histology, n (%)			0.52 ƒ
Infiltrating ductal carcinoma	25 (64.1%)	6 (75%)	
Infiltrating canalicular carcinoma	9 (23.1%)	1 (12.5%)	
Infiltrating lobular carcinoma	5 (12.8%)	1 (12.5%)	
T, n (%)			0.52 ƒ
1	1 (2.5%)	1 (11.1%)	
2	15 (37.5%)	4 (44.4%)	
3	15 (37.5%)	3 (33.3%)	
4	9 (22.5%)	1 (11.1%)	
N, n (%)			
0	0	2 (22.2%)	0.05 ƒ
1	18 (45%)	2 (22.2%)	
2	12 (30%)	3 (33.3%)	
3	10 (25%)	2 (22.2%)	
Clinical stage, n (%)			0.47 ƒ
IIB	8 (20%)	4 (44.4%)	
IIIA	16 (40%)	3 (33.3%)	
IIIB	5 (12.5%)	0	
IIIC	11 (27.5%)	2 (22.2%)	
Ki67 expression %, median (range)	30 (20–40)	30 (20–40)	0.73 §

## Data Availability

For breast cancer cell lines analysis are listed in the Appendix A.

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
