# Peer review of "The Expression Profiles of lncRNAs Are Associated with Neoadjuvant Chemotherapy Resistance in Locally Advanced, Luminal B-Type Breast Cancer"

_ijms, 2024, doi:10.3390/ijms25158077_

Round 1

Reviewer 1 Report

Comments and Suggestions for Authors

The article titled "The expression profile of lncRNAs is associated with neoadjuvant chemotherapy resistance in locally advanced, luminal B-type breast cancer" by Miguel González-Woge et al. provides valuable insights. However, there are several areas that require attention:

1. The authors analyzed lncRNA expression profiles from breast cancer cell lines and patient tumor samples but did not include normal breast tissue samples for comparison.

2. The authors did not perform validation of the identified lncRNA signature in an independent cohort of breast cancer patients.

3. The authors only focused on luminal breast cancer subtypes and did not evaluate the potential diagnostic and predictive value of the identified lncRNA signature in other breast cancer subtypes.

4. The present study relied on RNA-Seq data from cell lines and patient tumor samples. However, it does not capture the complex and dynamic nature of the tumor microenvironment, which could influence the expression of lncRNAs.

5. This study only focused on a limited number of lncRNAs and did not explore the potential interactions or functional roles of these lncRNAs in luminal breast cancer.

6. The authors could include the mechanisms by which lncRNAs contribute to luminal phenotypes and neoadjuvant chemotherapy resistance that may provide important insights into their clinical significance.

7. For statistical analysis, multivariate regression models would be needed to evaluate the independent predictive value of GATA3-AS1 and account for potential confounding factors.

8. Additional validation studies related to GATA3-AS1 using independent cohorts are necessary to establish its robustness and clinical utility.

Comments on the Quality of English Language

Moderate editing of English language required

Author Response

For research article

Response to Reviewer 1 Comments

1. Summary

Thank you very much for taking the time to review this manuscript. Please find the detailed responses below and the corresponding corrections highlighted in the re-submitted files.

2. Questions for General Evaluation

Reviewer’s Evaluation

Response and Revisions

Does the introduction provide sufficient background and include all relevant references?

Yes/Can be improved/Must be improved/Not applicable

All comments were analyzed and resolved with proper modifications highlighted in the manuscript.

Are all the cited references relevant to the research?

Yes/Can be improved/Must be improved/Not applicable

Is the research design appropriate?

Yes/Can be improved/Must be improved/Not applicable

Are the methods adequately described?

Yes/Can be improved/Must be improved/Not applicable

Are the results clearly presented?

Yes/Can be improved/Must be improved/Not applicable

Are the conclusions supported by the results?

Yes/Can be improved/Must be improved/Not applicable

3. Point-by-point response to Comments and Suggestions for Authors

Comments 1: The authors analyzed lncRNA expression profiles from breast cancer cell lines and patient tumor samples but did not include normal breast tissue samples for comparison.

Response 1: Thank you for pointing this out. Our study is focused on the identification of lncRNAs as predictive molecular biomarkers for neoadjuvant chemotherapy response. Thus, these lncRNAs are proposed to be used when the patient has already been diagnosed with breast cancer, in order to support the treatment scheme decision, identifying those patients that will not have benefit from neoadjuvant chemotherapy and is not an aim of this study to analyzed their diagnostic application. Then, we design our study as a case-control analysis, in which our control group are the patients who response to neoadjuvant chemotherapy, and normal samples are not considered in the study design, as it has been established for similar studies (https://doi.org/10.1016/j.ccell.2022.05.005).

Comments 2: The authors did not perform validation of the identified lncRNA signature in an independent cohort of breast cancer patients.

Response 2: We agree with you comment, thank you. To resolve this point, we added to our study a differential expression gene analysis from an independent cohort of RNA-Seq Data of 20 Luminal B patients, as you can see in figure 7, in which we added panels C and D (line 351), that contains the overexpression validation of GATA3-AS1 and MAPT-IT1.

Comments 3: The authors only focused on luminal breast cancer subtypes and did not evaluate the potential diagnostic and predictive value of the identified lncRNA signature in other breast cancer subtypes.

Response 3: Thank you for pointing this out. We focused this study on the identification of molecular biomarkers in luminal subtypes, particularly in Luminal B patients, because it is the most common molecular subtypes in breast cancer diagnosis (DOI: 10.3322/caac.21583). Although they are the most frequent diagnostic in clinic, luminal B patients have no proper clinical definition and this leads to inadequate treatment selection (doi: 10.31557/APJCP.2019.20.8.2247). Thus, luminal patients, particularly luminal B patients, are a group of clinical interest in breast cancer clinic that need further research for predictive molecular biomarkers identification. To highlight this information in our manuscript, we modified Discussion section, in line 458.

Comments 4: The present study relied on RNA-Seq data from cell lines and patient tumor samples. However, it does not capture the complex and dynamic nature of the tumor microenvironment, which could influence the expression of lncRNAs.

Response 4: We agree with your comment, thank you. We agree that RNA-Seq from bulk samples does not allow the characterization of the breast tumor microenvironment, which has important implications in tumor development, response to treatment and clinical outcome. However, the analysis of transcriptomes from bulk samples has lead to the development of molecular biomarkers, such as Oncotype Dx (DOI: 10.1056/NEJMoa041588), since it allows the analysis of differentially expressed genes considering the whole tumor organization and structure, which is the most common approach in clinical practice. Thus, we added as perspectives the spatial transcriptomics importance in the identification of novel molecular biomarkers in Discussion section, in line 599.

Comments 5: This study only focused on a limited number of lncRNAs and did not explore the potential interactions or functional roles of these lncRNAs in luminal breast cancer.

Response 5: Thank you for pointing this out. Our study design, as is describedin the manuscript, is focused on the identification of predictive molecular biomarkers based on lncRNAs for luminal B breast cancer patients that are candidates to receive neoadjuvant chemotherapy. Thus, we initiate our screening identifying lncRNAs associated to luminal phenotype in breast cancer cell lines, and then in breast cancer tumor samples (from GSE123845 and GSE163882 cohorts, as described in line 290), looking for those lncRNAs that are differentially expressed in both luminal breast cancer cell lines and luminal breast cancer patients (as is shown in figure 6), that then were validated in two independent cohorts (as described in line 306). Once we identified several candidates, we performed and independent differential expression gene analysis in GSE123845 and GSE163882 cohorts samples, in which we added as a variable of experiment design the response to neoadjuvant chemotherapy, as described in line 338, in order to identify the lncRNAs associated to the response to neoadjuvant chemotherapy (we noticed this was not clearly explained in the manuscript, thus we made a correction in line 340). Our results showed that the only lncRNA that is consistently associated to luminal phenotype and to neoadjuvant chemotherapy response was GATA3-AS1, which we then validated by RNA-ISH and RT-qPCR. Additionally, we agree that the analysis of the potential functional roles of the lncRNAs identified in this study is necessary to understand the biological functions underlying the clinical utility of these lncRNAs as predictive molecular biomarkers to neoadjuvant chemotherapy response. In order to enrich our work, we included the gene set enrichment analysis associated to the differentially expressed lncRNAs, modified in line 369

Comments 6: The authors could include the mechanisms by which lncRNAs contribute to luminal phenotypes and neoadjuvant chemotherapy resistance that may provide important insights into their clinical significance.

Response 6: We agree with your comment. We included the gene set enrichment analysis associated to the differentially expressed lncRNAs in supplemental figure 4.

Comments 7: For statistical analysis, multivariate regression models would be needed to evaluate the independent predictive value of GATA3-AS1 and account for potential confounding factors.

Response 7: Thank you for pointing this out. We performed the multivariate regression and we found that GATA3-AS1 expression is an independent predictor for residual disease in luminal B breast cancer patients treated with neoadjuvant chemotherapy (odds ratio = 1.48, p value = 0.024). We added this information to the manuscript in line 458.

Comments 8: Additional validation studies related to GATA3-AS1 using independent cohorts are necessary to establish its robustness and clinical utility.

Response 8: Thank you for your comment. To validate our results, we added a differential expression gene analysis from an independent cohort of RNA-Seq Data of 20 Luminal B patients, as you can see in figure 7, in which we added panels C and D (line 340), that contains the overexpression validation of GATA3-AS1 associated to resistance to neoadjuvant chemotherapy.

4. Response to Comments on the Quality of English Language

Point 1:  Not applicable

Response 1:    (in red)

5. Additional clarifications

We like to notice that this manuscript was submitted for english edition (D53D-58EE-1DFB-42B7-92AP) by American Journal Experts.

Reviewer 2 Report

Comments and Suggestions for Authors

Comments:

1. The author needs to rephrase the first line of the abstract. The present line means that LncRNAs are expressed only in breast cancer. 

2. In the line "overall survival of the pCR 73 population of Her2," please make it Her2 "positive."

3. What are these "45 transcripts with other functions were downregulated"?. what are these 1,546 mRNAs and 499 155 lncRNAs other than 585 mRNAs, 199 lncRNAs?

4. "from the 49 BC cell lines." Is that 42? Not 49.

5. How did you perform molecular subtyping in the BC cell line? 

6. The heatmap in Figs 2A and 5 may be according to BC subtypes. The clustering should be according to subtypes. LA and LB, Her-2+/TNBC, are mixing up in the present setting. There may also be normal samples. 

Author Response

For research article

Response to Reviewer 2 Comments

1. Summary

Thank you very much for taking the time to review this manuscript. Please find the detailed responses below and the corresponding corrections highlighted in the re-submitted files.

2. Questions for General Evaluation

Reviewer’s Evaluation

Response and Revisions

Does the introduction provide sufficient background and include all relevant references?

Yes/Can be improved/Must be improved/Not applicable

All comments were analyzed and resolved with proper modifications highlighted in the manuscript.

Are all the cited references relevant to the research?

Yes/Can be improved/Must be improved/Not applicable

Is the research design appropriate?

Yes/Can be improved/Must be improved/Not applicable

Are the methods adequately described?

Yes/Can be improved/Must be improved/Not applicable

Are the results clearly presented?

Yes/Can be improved/Must be improved/Not applicable

Are the conclusions supported by the results?

Yes/Can be improved/Must be improved/Not applicable

3. Point-by-point response to Comments and Suggestions for Authors

Comments 1: The author needs to rephrase the first line of the abstract. The present line means that LncRNAs are expressed only in breast cancer.

Response 1: We agree with your observation. Proper modifications have been made in the abstract (line 29).

Comments 2: In the line "overall survival of the pCR 73 population of Her2," please make it Her2 "positive."

Response 2: Thank you for your observation. It has been corrected in line 76 according to your comment.

Comments 3: What are these "45 transcripts with other functions were downregulated"?. what are these 1,546 mRNAs and 499 155 lncRNAs other than 585 mRNAs, 199 lncRNAs?

Response 3: Thank you for your observation. The transcripts annotated as “Other” in our analysis are principally pseudogenes and snoRNAs, according with the Ensembl gene IDs and gene classification (release 112). To answer your comment, we performed a gene set enrichment analysis with the 2874 differentially expressed that you pointed out, and we found that these genes are related with cell migration, cell differentiation and extracellular features, as it is shown in supplemental figure 2.

Comments 4: "from the 49 BC cell lines." Is that 42? Not 49.

Response 4: Thank you for your observation. It has been corrected in line 158 according to your comment.

Comments 5: How did you perform molecular subtyping in the BC cell line?

Response 5: We appreciate you point this out. We considered the molecular subtype specified for each cell line accordingly to the description included in the GEO datasets sample information. Additionally, we also performed a transcriptome similarity with clustering analysis for 151 breast cancer cell lines, that showed five groups of breast cancer cell lines, accordingly with the breast cancer subtype associated in their sample description. From these analysis, we selected 42 breast cancer cell lines, taking into account their clustering in each subtype group. In order to resolve your comment, we added a supplemental figure (Supplemental Figure 1) to our manuscript with the information given above.

Comments 6: The heatmap in Figs 2A and 5 may be according to BC subtypes. The clustering should be according to subtypes. LA and LB, Her-2+/TNBC, are mixing up in the present setting. There may also be normal samples.

Response 6: Thank you for your comment. Breast tumors are characterized for their molecular and biological heterogeneities, which make them of clinical interest in oncology research (https://doi.org/10.1038/s41416-021-01328-7). This heterogeneity is also identified in the transcriptome profile of the samples analyzed in this study, as is shown in figures 2, 3 and 5 in the manuscript, and since our approach to analyze transcriptome profiles integrates hierarchical clustering, the consequence is that the clustering of the samples by their transcriptome profile has not an exact overlapping with the molecular subtype reported for each sample, as it has been described before in scientific literature (https://doi.org/10.1016/j.ccell.2018.03.014). However, as it is shown in Figures 2 and 5, there are defined clusters which integrate luminal A and B samples (orange and red samples) and HER-enriched and basal samples (green and blue samples), which is due to their transcriptional profile similarities, as it has been described before by Perou and collaborators (https://doi.org/10.1038/35021093). Finally, for the construction of figure 2, we performed initially the transcriptome profile analysis of 150 breast cancer cell lines, which is shown in supplemental figure 1, and in this analysis we included transformed breast cell lines (normal control). Since our aim was to identify lncRNAs related to luminal phenotype, we constructed as main figure 2 the heatmap only with the 42 breast cancer cell lines indicated in the manuscript. For figure 5, the GSE123845 and GSE163882 cohorts included normal samples, which are highlighted in yellow in the heatmap annotations.

4. Response to Comments on the Quality of English Language

Point 1:

Response 1:   Not Applicable

5. Additional clarifications

We like to notice that this manuscript was submitted for english edition (D53D-58EE-1DFB-42B7-92AP) by American Journal Experts.

Reviewer 3 Report

Comments and Suggestions for Authors

This study identified a lncRNA signature specific for luminal breast cancer, especially for the luminal B subtype. The luminal-specific lncRNAs were identified using 42 BC cell lines as well as 320 BC patient tumor samples, and then validated by RNA-ISH experiments. The manuscript is generally well-written. I only have some minor concerns as below.

Introduction, line 69: Typo for “according”.

Introduction, line 75: A reference is needed for “It is also known that compared with luminal A, luminal B is nearly thirty times more likely to achieve pCR”.

Results, line 153: The differential expression gene analysis in cell lines was not clearly described. Were the 5 normal cell lines used as the reference? How many genes and lncRNAs genes were included in the test? What was the significant threshold used to adjust for the test burden? Was the number of subtypes also adjusted for the test burden? This key information should be provided here since the detailed methods section was placed at the end of the manuscript.

Results, line 162: How were the top 40 lncRNAs ranked? Based on p-values or fold of change?

Results, line 169: “Differential expression analysis was performed to assess the expression profiles of lncRNAs associated with the luminal subtype B, which revealed that 1,182 mRNAs, 157 transcripts 197 with other functions, and 397 lncRNAs”. It first described the differential expression analysis was performed for lncRNAs only, but then the results included mRNA and lncRNA. Multiple results and figures (e.g. volcano plots) in this study showed that the mRNAs were included in the analyses. However, only the lncRNAs were focused and discussed. If this study only focused on lncRNA, it should provide a reason and should not include the mRNA in the analysis. Otherwise, the mRNA results need to be discussed as long as they were included in the analyses.

Figure 1: The figure 1 did not clearly show where the 42 BC cell lines and the 320 BC tumors were used separately.

Discussion: 320 malignant breast tumors were from two independent cohorts: a Korean cohort and a US cohort. There could be potential differences in the genetic regulation between Asian and European ancestries. This heterogeneity should be mentioned in the discussion.

Author Response

For research article

Response to Reviewer 3 Comments

1. Summary

Thank you very much for taking the time to review this manuscript. Please find the detailed responses below and the corresponding corrections highlighted in the re-submitted files.

2. Questions for General Evaluation

Reviewer’s Evaluation

Response and Revisions

Does the introduction provide sufficient background and include all relevant references?

Yes/Can be improved/Must be improved/Not applicable

All comments were analyzed and resolved with proper modifications highlighted in the manuscript.

Are all the cited references relevant to the research?

Yes/Can be improved/Must be improved/Not applicable

Is the research design appropriate?

Yes/Can be improved/Must be improved/Not applicable

Are the methods adequately described?

Yes/Can be improved/Must be improved/Not applicable

Are the results clearly presented?

Yes/Can be improved/Must be improved/Not applicable

Are the conclusions supported by the results?

Yes/Can be improved/Must be improved/Not applicable

3. Point-by-point response to Comments and Suggestions for Authors

Comments 1: Introduction, line 69: Typo for “according”..

Response 1: Thank you for your observation. It has been corrected in line 71 according to your comment.

Comments 2: Introduction, line 75: A reference is needed for “It is also known that compared with luminal A, luminal B is nearly thirty times more likely to achieve pCR”.

Response 2: Thank you for your observation. It has been corrected in line 78 according to your comment.

Comments 3: Results, line 153: The differential expression gene analysis in cell lines was not clearly described. Were the 5 normal cell lines used as the reference? How many genes and lncRNAs genes were included in the test? What was the significant threshold used to adjust for the test burden? Was the number of subtypes also adjusted for the test burden? This key information should be provided here since the detailed methods section was placed at the end of the manuscript.

Response 3: Thank you for pointing this out. We performed our analysis considering transformed cell lines (normal cell lines) as the reference. All transcripts with >1 count were included in the differential gene analysis. We added these specifications to the results section of the paper in addition to the methods section. On the other hand, the molecular subtypes were considered as the main variables in the DESeq2 differential gene expression model. We also used Bonferroni’s correction to address possible false positive results. Adjusted P values <0.01 were considered as significant.

Comments 4: Results, line 162: How were the top 40 lncRNAs ranked? Based on p-values or fold of change?

Response 4: Thank you for your comment. The top 40 lncRNAs were ranked based on the variance values associated to the subtype variable, and the complete list of these lncRNAs are included in Supplemental Table 1.

Comments 5: Results, line 169: “Differential expression analysis was performed to assess the expression profiles of lncRNAs associated with the luminal subtype B, which revealed that 1,182 mRNAs, 157 transcripts 197 with other functions, and 397 lncRNAs”. It first described the differential expression analysis was performed for lncRNAs only, but then the results included mRNA and lncRNA. Multiple results and figures (e.g. volcano plots) in this study showed that the mRNAs were included in the analyses. However, only the lncRNAs were focused and discussed. If this study only focused on lncRNA, it should provide a reason and should not include the mRNA in the analysis. Otherwise, the mRNA results need to be discussed as long as they were included in the analyses.

Response 5: Thank you for pointing this out. We included the supplemental figure 2, which integrates the gene set enrichment analysis of mRNAs and lncRNAs differentially expressed in breast cancer cell lines, which includes those  1,182 mRNAs, 157 transcripts 197 with other functions, and 397 lncRNAs mentioned in the manuscript. In this analysis we found that these differentially expressed genes showed their association with biological processed involved in mammary tumor development, such as cell differentiation, cell migration and extracellular matrix organization, as is mentioned in line 163. Moreover, we also included an analysis of differentially expressed of mRNAs and lncRNAs differentially expressed in the patients from GSE123845 and GSE163882 cohorts (supplemental figure 3), in which we found that those mRNAs and lncRNAs differentially expressed are related to  biological processes related to neuronal processes, such as glutamaergic and serotonergic synapses, as well as pathways related to hormone response, such as estrogen signaling pathway, as is mentioned in the manuscript, in line 369.

Comments 6: Figure 1: The figure 1 did not clearly show where the 42 BC cell lines and the 320 BC tumors were used separately.

Response 6: Thank you for your observation. Figure 1 has been modified according to your comment and also the figure description (line 144).

Comments 7: Discussion: 320 malignant breast tumors were from two independent cohorts: a Korean cohort and a US cohort. There could be potential differences in the genetic regulation between Asian and European ancestries. This heterogeneity should be mentioned in the discussion.

Response 7: We appreciate your observation, and we agree with your comment. Tumor biology heterogeneity due to race and ethnicity was added to Discussion section (line 542).

4. Response to Comments on the Quality of English Language

Point 1:

Response 1:   Not Applicable.

5. Additional clarifications

We like to notice that this manuscript was submitted for english edition (D53D-58EE-1DFB-42B7-92AP) by American Journal Experts.

Round 2

Reviewer 1 Report

Comments and Suggestions for Authors

Accept in present form

Comments on the Quality of English Language

 Moderate editing of English language required

Reviewer 2 Report

Comments and Suggestions for Authors

The authors have included all the changes.